## TRANSLATIONAL PERSPECTIVE

# Tendon's 'first aid' response – new insights into acute tendon molecular reprogramming after rupture

M. V. Franchi[1] , C. S. Fry[2] and M. Kjaer[3]

[1]*Neuromuscular Physiology Lab, Department of Biomedical Sciences, University of Padova, Padova, Italy*
[2]*Center for Muscle Biology, University of Kentucky, Lexington, Kentucky, USA*
[3]*Institute of Sports Medicine, Copenhagen University Hospital – Bispebjerg–Frederiksberg, Copenhagen, Denmark*

Email: Martino.franchi@unipd.it

Handling Editors: Karyn Hamilton & Ken O'Halloran

The peer review history is available in the Supporting Information section of this article (https://doi.org/10.1113/JP289048#support-information-section).

Tendon injuries represent a significant challenge in orthopaedic medicine, with far-reaching implications for patient mobility, quality of life and healthcare resources. Acute tendon ruptures, often occurring against a backdrop of pre-existing tendinopathy, typically necessitate surgical intervention or extended immobilization, leading to prolonged recovery periods and potential long-term complications.

The healing process of tendons is notoriously problematic, frequently resulting in the formation of weaker tissue that fails to match the mechanical, structural and material properties of healthy tendon extracellular matrix (ECM). This suboptimal healing not only may impede patients' return to pre-injury activity levels but also increases the risk of re-rupture and progression to chronic tendon disease.

Despite the prevalence and impact of tendon injuries, our understanding of the cellular mechanisms underlying tendon homeostasis and early injury response has been limited. This knowledge gap has hindered the development of effective biological interventions and early clinical rehabilitation strategies to enhance tendon repair. However, recent advancements in single-cell transcriptomic mapping of tendons have begun to illuminate the complex cellular and molecular landscapes of healthy human tendons, offering new avenues for research and potential therapeutic targets.

The recent publication by Mimpen et al. (2025) provides valuable insights into the microenvironmental changes occurring in ruptured tendons; by exploiting single-nucleus RNA sequencing and comprehensive transcriptomic analyses, the study describes a predominant role of fibroblasts and endothelial cells in the coordination of an early injury response in ruptured human tendon. While gene expression does not guarantee phenotypic adaptations, the findings by Mimpen et al. reveal an increase in fibroblast subsets associated with ECM deposition and fibrosis. This provides new insight into the shift from relatively dormant cell activity in healthy tendons to a reparative state. These observations offer valuable clues about changes in tendon gene activity during the transition from homeostasis to repair, despite the complex relationship often observed between gene expression and final protein production.

Mimpen and colleagues' study also effectively demonstrates angiogenesis as an integral part of tendon reparation. Interestingly, a recent investigation of human ruptured Achilles tendons found no increase in cumulative collagen formation (i.e. measured by deuterium oxide tracing techniques) within the first 2 weeks after rupture, suggesting that enhanced collagen production is not an immediate response to injury (Cramer et al., 2023). This finding, when considered alongside the current publication, raises important questions about the relationship between increased expression of genes related to ECM organization and cell cycle signalling, and their regulation of *in vivo* protein synthesis in ruptured and healing human tendons. Possibly, one of the caveats of Mimpen and colleagues' study is represented by the older age of the volunteers with tendon rupture (∼70 years old) compared to their healthy controls (∼36 years old) or to those investigated by Cramer et al. (∼44 years old) (Cramer et al., 2023). The mechanisms governing this interplay remain unresolved and warrant further investigation.

The current study affords insight into changes in the microenvironment of ruptured tendons, demonstrating increases in the expression of genes related to collagen organization and biomineralization (Mimpen et al., 2025). Calcium deposits resulting from biomineralization can result in calcific tendinopathy, compromising structural integrity, and increasing the risk of rupture. These processes most commonly affect the supraspinatus tendon of the shoulder and are rarely described within the quadriceps tendon (Abram et al., 2012; Oliva et al., 2012). Even though the expression of biomineralization pathways was increased, it is known that enhanced gene expression does not always translate to phenotypic changes. However, while far from conclusive, the single nucleus and pseudobulk analyses presented by Mimpen and colleagues still provide unique observations surrounding mineralization pathogenesis that may underlie structural tendon changes that predispose to injury.

The cellular response in ruptured tendons plays a vital role in healing, and the display of its immediate reaction in human tendon as shown in the present publication is significant (Mimpen et al., 2025). However, it is also important to note that the mechanical loading applied on the tendon during rehabilitation plays a vital role as, for example, tendons can experience a surprisingly quick elongation even after a week from surgery, and this can impair the long-term muscular tendinous function (Hoeffner et al., 2024).

The investigation of human tendon has until now been limited by the ability to sample human tissue, and this has in many ways limited the insight we so far have obtained in several connective tissues such as ligament, tendon and cartilage, as compared to the major mechanistic insight we have in skeletal muscle due to, for example, repeated biopsy sampling, which is not possible in tendon without invalidating the results (Heinemeier et al., 2016). The authors demonstrate creativity in their sampling approach, utilizing both ruptured and healthy tendons from different surgical contexts. Despite challenges such as a small sample size and significant age disparity between groups, the study's strength lies in obtaining samples from the same tendon types in both conditions. This innovative design, leveraging traumatic events for open tissue sampling and employing percutaneous sampling in healthy tendons, exemplifies

the necessary approach to advance our understanding of human tendon processes. Thus, the study serves as an excellent model for future research in this challenging field.

The identification of specific cell types and gene expression profiles in ruptured tendons, consistent with findings from other injuries, tendinopathic specimens and fibrotic pathologies, provides a foundation for understanding the complex cellular responses in tendon healing, as functional characterization of fibroblasts as key orchestrators of early injury response may reveal potential targets for modulating long-term repair outcomes.

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

## Additional information

### Competing interests

None of the authors have any conflicts of interest.

### Author contributions

M.F.: Conception or design of the work; Drafting the work or revising it critically for important intellectual content; Final approval of the version to be published; Agreement to be accountable for all aspects of the work. C.F.: Conception or design of the work; Drafting the work or revising it critically for important intellectual content; Final approval of the version to be published; Agreement to be accountable for all aspects of the work. M.K.: Conception or design of the work; Drafting the work or revising it critically for important intellectual content; Final approval of the version to be published; Agreement to be accountable for all aspects of the work.

### Funding

No funding was received.

### Acknowledgements

Open access publishing facilitated by Universita degli Studi di Padova, as part of the Wiley - CRUI-CARE agreement.

### Keywords

tendon, tendon acute rupture, transcriptomics

### Supporting information

Additional supporting information can be found online in the Supporting Information section at the end of the HTML view of the article. Supporting information files available:

**Peer Review History**

