## [Peer Review History · The Journal of Physiology]

Tendon's "first aid" response – new insights into acute tendon molecular reprogramming after rupture

Martino V. Franchi, Christopher S Fry, and Michael Kjaer

DOI: 10.1113/JP289048

Corresponding author(s): Martino Franchi (Martino.franchi@unipd.it)

Review Timeline:

Submission Date:	22-May-2025
Editorial Decision:	09-Jun-2025
Revision Received:	17-Jun-2025
Accepted:	30-Jun-2025

Senior Editor: Karyn Hamilton

Reviewing Editor: Ken O'Halloran

Transaction Report:

Dear Dr Franchi,

Re: JP-TP-2025-289048 "**Tendon's "first aid" response - new insights into acute tendon molecular reprogramming after rupture**" by Martino V. Franchi, Christopher S Fry, and Michael Kjaer

Thank you for submitting your manuscript to The Journal of Physiology. It has been assessed by a Reviewing Editor and by 1 expert referee and we are pleased to tell you that it is acceptable for publication following minor revision.

The review comments are copied at the end of this email.

Please address all the points raised and incorporate all requested revisions or explain in your Response to Referees why a change has not been made. We hope you will find the comments helpful and that you will be able to return your revised manuscript within 2 weeks. If you require longer than this, please contact journal staff: jp@physoc.org.

REVISION CHECKLIST: Upload a full Response to Referees file. To create your 'Response to Referees' copy all the reports, including any comments from the Senior and Reviewing Editors, into a Microsoft Word, or similar, file and respond to each point, using font or background colour to distinguish comments and responses and upload as the required file type.

- 'Potential Cover Art' for consideration as the issue's cover image
- Appropriate Supporting Information (video, audio or data set: see https://jp.msubmit.net/cgi-bin/main.plex?form_type=display_requirements#supp).

We look forward to receiving your revised submission.

Yours sincerely,

Karyn Hamilton
Senior Editor
The Journal of Physiology

EDITOR COMMENTS

Reviewing Editor:

Comments to the Author:

Thank you for providing this perspective article linked to the original study by Mimpfen et al. 2025. The referee has provided some suggestions requiring minor edits. I feel you have already emphasized that gene expression does not always translate to phenotypic change but please consider the referee's point concerning calcification. Also, please revise lines 61-63, which contain grammatical errors disrupting the flow of the narrative.

Senior Editor:

Comments to the Author:

Thank you for the time and effort invested in this perspective. It will be a nice addition to the original research and appreciated by our readership. At this point, I'd like to provisionally accept it pending the minor revisions suggested by the Referee. We look forward to seeing the revised work. Thank you again!

REFeree COMMENTS

Referee #1:

We would like to thank the authors for the time and effort they put into highlighting our paper and for nicely summarising our findings.

We have a few minor suggestions for the authors:

- Sentence 49: missing reference (REF).
- Sentence 52: Reference? What does (1) refer to?
- Sentence 68: authors of "the" present paper
- Sentences 68-71: the two sentences seem to make the same point.
- We advise the authors to use of the word "reparative" instead of "regenerative"
- The authors highlight the potential implication of mineralisation. The mineralisation pathways were found to be increased in terms of gene expression levels, but osteoblasts were found mainly in one sample. Therefore, we advise the authors to use caution with the interpretation of this finding as activation of the pathway does not necessarily mean that mineralisation will take place.

END OF COMMENTS

Dear Editors,
Dear Reviewer,

We would like to thank you for the comments and advice made in relation to our translational perspective. We have taken into consideration all your suggestions and I have listed them below in a "point-by-point" response fashion.

EDITOR COMMENTS

Reviewing Editor:

Comments to the Author:

Thank you for providing this perspective article linked to the original study by Mimpfen et al. 2025. The referee has provided some suggestions requiring minor edits. I feel you have already emphasized that gene expression does not always translate to phenotypic change but please consider the referee's point concerning calcification. Also, please revise lines 61-63, which contain grammatical errors disrupting the flow of the narrative.

As the Reviewing Editor stated, we have already emphasised that gene expression does not always translate to protein content change. We have decided anyway to thoroughly consider the comments of Reviewer 1 and rephrased the sentence (reported below in the specific responses to Rev 1)

Line 61-63 have also been amended, and now they read:

"However, it is also important to note that the mechanical loading applied on the tendon during rehabilitation plays a vital role as, for example, tendons can experience a surprisingly quick elongation even after a week from surgery, and this can impair the long-term muscular tendinous function"

Senior Editor:

Comments to the Author:

Thank you for the time and effort invested in this perspective. It will be a nice addition to the original research and appreciated by our readership. At this point, I'd like to provisionally accept it pending the minor revisions suggested by the Referee. We look forward to seeing the revised work. Thank you again!

Thank you for your kind words, Dr Hamilton!

REFeree COMMENTS

Referee #1:

We would like to thank the authors for the time and effort they put into highlighting our paper and for nicely summarising our findings.

We would like to thank the authors for a very nice article! It captured our attention.

We have a few minor suggestions for the authors:

- Sentence 49: missing reference (REF).

Amended as requested.

- Sentence 52: Reference? What does (1) refer to?

Amended as requested. We referred to Mimpfen et al. study.

- Sentence 68: authors of "the" present paper

Amended as requested.

- Sentences 68-71: the two sentences seem to make the same point.

Thank you for spotting this! One of the sentences belonged to a previous draft. Amended as requested.

- We advise the authors to use of the word "reparative" instead of "regenerative"

Amended as requested.

- The authors highlight the potential implication of mineralisation. The mineralisation pathways were found to be increased in terms of gene expression levels, but osteoblasts were found mainly in one sample. Therefore, we advise the authors to use caution with the interpretation of this finding as activation of the pathway does not necessarily mean that mineralisation will take place.

We agree with the Reviewer's comment. We now have taken into consideration this comment together with the one from the Reviewing Editor and we have rephrased the specific section:

"Even though the expression of biomineralization pathways was increased, it is known that enhanced gene expression does not always translate to phenotypic changes. However, while far from conclusive, the single nucleus and pseudobulk analyses presented by Mimpfen and colleagues still provide unique observations surrounding mineralization pathogenesis that may underlie structural tendon changes that predispose to injury."

Dear Dr Franchi,

Re: JP-TP-2025-289048R1 "**Tendon's "first aid" response - new insights into acute tendon molecular reprogramming after rupture**" by Martino V. Franchi, Christopher S Fry, and Michael Kjaer

We are pleased to tell you that your paper has been accepted for publication in The Journal of Physiology.

Authors should note that it is too late at this point to offer corrections prior to proofing. Major corrections at proof stage, such as changes to figures, will be referred to the Editors for approval before they can be incorporated. Only minor changes, such as to style and consistency, should be made at proof stage. Changes that need to be made after proof stage will usually require a formal correction notice.

If you would like to receive our 'Research Roundup', a monthly newsletter highlighting the cutting-edge research published in The Physiological Society's family of journals (The Journal of Physiology, Experimental Physiology and Physiological Reports), please click this link, fill in your name and email address and select 'Research Roundup':
<https://www.physoc.org/journals-and-media/membernews/>

Yours sincerely,

Karyn Hamilton
Senior Editor
The Journal of Physiology

EDITOR COMMENTS

Reviewing Editor:

Comments to the Author:

Thank you for your responses and for making revisions to the text. The Translational Perspective provides a nice companion article to the original research article. We are grateful to you for your time and insights shared in the article.

Senior Editor:

Comments to the Author:

Thank you for these revisions and for your efforts on this manuscript. We are pleased to accept it for publication in The Journal of Physiology!

P.S. - You can help your research get the attention it deserves! Check out Wiley's free Promotion Guide for best-practice recommendations for promoting your work at www.wileyauthors.com/eoo/guide. You can learn more about Wiley Editing Services which offers professional video, design, and writing services to create shareable video abstracts, infographics, conference posters, lay summaries, and research news stories for your research at www.wileyauthors.com/eoo/promotion.

IMPORTANT NOTICE ABOUT OPEN ACCESS: To assist authors whose funding agencies mandate public access to published research findings sooner than 12 months after publication, The Journal of Physiology allows authors to pay an Open Access (OA) fee to have their papers made freely available immediately on publication.

You can check if your funder or institution has a Wiley Open Access Account here: <https://authorservices.wiley.com/author-resources/Journal-Authors/licensing-and-open-access/open-access/author-compliance-tool.html>.